# Mechanical properties of a polylactic 3D-printed interim crown after thermocycling

**Re-Mee Doh[1], Won-Il Choi[2], Seo Young Kim [2], Bock-Young Jung [2]***

1 Department of Advanced General Dentistry, College of Dentistry, Dankook University, Cheonan, Korea,
2 Department of Advanced General Dentistry, College of Dentistry, Yonsei University, Seoul, Korea

* JBY1004@yuhs.ac

**Data Availability Statement:** All relevant data are within the manuscript and its Supporting Information files.

**Funding:** This work was supported by a faculty research grant from the Yonsei University College

## Abstract

Polylactic acid (PLA) has garnered attention for use in interim dental restorations due to its biocompatibility, biodegradability, low cost, ease of fabrication, and moderate strength. However, its performance under intraoral conditions, particularly under heat and moisture, remains underexplored. This study evaluated the mechanical properties of PLA interim crowns compared with those of polymethylmethacrylate (PMMA) and bisphenol crowns under simulated intraoral conditions with thermocycling. Three CAD/CAM polymers—PMMA (milling), PLA (fused deposition), and bisphenol (stereolithography)—were tested for fracture resistance, hardness, and surface roughness. For fracture strength, 25 crowns from each group were cemented onto dies. The Shore D hardness and surface roughness were measured on round discs before and after 10,000 thermocycles (5˚C/55˚C). The surface topography was assessed via scanning electron microscopy. PMMA exhibited the highest fracture strength (2787.93 N), followed by bisphenol (2165.47 N) and PLA (2088.78 N), with no significant difference between the latter two. PMMA and bisphenol showed vertical fractures and cracks, whereas PLA showed crown tearing or die deformation. Bisphenol had the highest Shore D hardness, followed by PMMA and PLA, with no significant changes after thermocycling. The surface roughness (Ra) was lowest for bisphenol and similar between PMMA and PLA. The roughness (Rz) increased from bisphenol to PMMA to PLA. The roughness of the PMMA remained unchanged after thermocycling, whereas the Ra but not the Rz of the PLA increased. Bisphenol showed a significant increase in both Ra and Rz (p<0.0001). In conclusion, PLA interim crowns demonstrated mechanical properties comparable to those of conventional PMMA and bisphenol crowns after thermocycling.

## Introduction

Interim restorations are an integral part of prosthodontic treatment. Interim restorations are used to protect the tooth structure until the final restoration is placed, maintain aesthetics and function during the healing period, evaluate patient acceptance and determine the feasibility of transitioning to the final restoration. For these purposes, appropriate physical properties, mechanical strength, color, ease of fabrication, retention of properties in the intraoral environment, and in vivo biocompatibility are essential [1].

of Dentistry (No. 6-2023-0015). The funders had no role in study design, data collection and analysis, decision to publish, or preparation of the manuscript.

**Competing interests:** The authors have declared that no competing interests exist.

Dental polymers primarily used as interim materials are limited to biomaterials such as polymethylmethacrylate (PMMA), bisphenol resins, and polyacrylate-ethene (PAEK). Although each material has certain limitations, such as high shrinkage of PMMA and low flowability of bisphenol A-glycidyl methacrylate (Bis-GMA) and urethane dimethacrylate (UDMA), they have been optimized for use in dental applications, minimizing the impact of these weaknesses [2].

As environmental issues such as material consumption and pollution are escalating, subtractive manufacturing technology is being replaced by additive manufacturing of restorations via various 3D printing technologies [3]. Compared with subtractive manufacturing, the additive method can reduce the consumption of material and energy and the wear of cutting tools. In addition, dental polymers for 3D printing are becoming increasingly diverse, and various types of materials, such as liquids, filaments, granules, and powders, are available for use with 3D printing technology, such as fused deposition modeling (FDM), stereolithography (SLA), digital light processing (DLP), and selective laser sintering (SLS) [4]. In FDM, products are fabricated via the extrusion of liquefied filaments or granules through a moving nozzle to layer materials on a scaffold. Compared with SLA and DLP, FDM has disadvantages, such as rougher surfaces and limited microrealization, but it is widely used for fabricating diagnostic models, customized individual trays, and provisional restorations in dentistry because of its advantages in terms of time and cost [4,5].

Unlike other dental polymers, such as polymethylmethacrylate (PMMA) and bisphenol resins, which contain residual monomers and elution additives that can cause cytotoxicity or systemic cytotoxicity [6–8], polylactic acid (PLA), derived from nontoxic natural renewable resources, is considered one of the most biocompatible and biodegradable biopolymers for use in suture materials, surgical membranes, medical implants, and orthopedic devices [9]. This suitability is due to the alpha-hydroxyl acids in PLA byproducts, which do not interfere with tissue healing and are excreted as water and carbon dioxide through the tricarboxylic acid cycle of the human body [10–13].

Owing to the advantages of PLA, such as its biocompatibility, biodegradability, ease of fabrication, moderate strength, low cost, and low energy demand for manufacturing, there has been interest in and research into the possibility of using PLA to produce interim restorations via additive manufacturing [14–16]. A few studies have investigated the possibility of using PLA in dental prostheses and reported a clinically acceptable marginal fit [3,16]. A study reported that a three-unit provisional fixed dental prosthesis (FDP) fabricated from PLA via FDM showed only deformation and not fracture because of its greater flexibility than the PMMA specimen fabricated via SLA or DLP [17]. Our previous study on the mechanical properties of PLA bar-shaped samples, in which intraoral conditions such as temperature and saliva were not considered, revealed that PLA FDM samples have lower flexural strength and surface roughness and a higher elastic modulus than milled PMMA and SLA-printed bisphenol samples do and that the mechanical properties of PLA FDM samples are within the clinically acceptable range [15]. However, few studies have investigated the clinical characteristics, including physiochemical and mechanical properties, in the intraoral environment. This information is important because an interim restoration should be able to withstand external stress from functional loads, saturated humidity, and changes in temperature for a period.

Therefore, this study aimed to evaluate the potential clinical use of PLA as a material for interim crowns by comparing its mechanical properties, such as fracture strength (FS), Shore D hardness, and surface roughness, with those of conventionally used CAD/CAM dental polymers, specifically PMMA (via subtractive manufacturing) and bisphenol (via SLA). Thermocycling was performed to replicate aging in the intraoral environment. The hypothesis was that

the mechanical properties of PMMA, bisphenol, and PLA would not significantly differ after thermocycling.

## Materials and methods

The PLA used in this study was approved as a material for interim crowns and bridges by the Korea Food and Drug Association (KFDA) after a series of tests, such as skin sensitization, intracutaneous reactivity, oral mucosa irritation, in vitro cytotoxicity, and dental device tests, which were conducted by the Yonsei University Medical Center.

### Test specimens and materials

Three types of conventional CAD/CAM polymers for interim restorations were tested. PMMA samples were fabricated via subtractive manufacturing. PLA samples were fabricated via FDM additive manufacturing, and bisphenol samples were fabricated via SLA manufacturing. The parameters applied to this investigation were the values that have been recommended in previous studies [17–19] or by the manufacturers, including the layer height, nozzle size, ejection speed, and curing time (Table 1). The bisphenol SLA samples were immersed in 100% isopropyl alcohol to remove the resin monomers (Medifive, Tornado, Korea), and postpolymerization was performed for 210 s by using a UV light polymerization unit (LC-3D print box, NextDent, Netherlands); however, the samples in the FDM group did not undergo postpolymerization processing.

For the interim single crown samples, a right first molar phantom tooth of the mandibular typodont system (Nissin Dental Product Inc., Tokyo, Japan) was prepared using 6° convergence and chamfer-ended margins, with an axial reduction of 1.5 mm and an occlusal reduction of 2.0 mm (Fig 1). A total of 75 dies of prepared teeth were made of POLYROCK (Cendres Metaux) through a conventional polyvinylsiloxane (PVS) impression procedure. Virtual images of the crown samples were obtained with a laboratory model scanner (Trios 4, 3Shape, Denmark). The samples were virtually designed with a CAD software program (exoCAD, exoCAD GmbH, Germany), converted into stereolithographic (STL) format, and then milled or 3D printed into crowns 8.0 mm in height with preset CAD parameters of 0.05 mm cement space and 0.05 mm above the margin line to be seated on the abutment [3,14] (Fig 1). The samples for each group were cemented on individual dies with Temp-bond E (Kerr, Brea, USA) under 50 N constant pressure for six minutes by one technician.

The samples used for the Shore D hardness and surface roughness tests were milled or 3D printed into round discs with a diameter of 5 mm and a thickness of 2 mm according to ISO 868 [20]. All of the crown samples were thermocycled for 10,000 cycles (5°C/55°C) with a

**Table 1. List of materials studied.**

| Group | Polymer | Manufacturer | Manufacturing method | Manufacturer |
|---|---|---|---|---|
| PLA | Polylactic acid | CUBICON Style–Plus-A15D CUBICON CO Korea | Fused deposition modeling, $\Phi = 1.75$ mm, Layer height: 0.1 mm, Nozzle: 0.4 mm, Speed: 60 m/s | KPLA, 3D KOREA CO Korea |
| PMMA | Polymethyl methacrylate | R2 solution MEGAGEN Korea | Subtractive milling | VIPI BLOCK TRILUX®, Vipi Odonto Products |
| Bisphenol | Bis-A ethoxylate dimethacrylate | KARV LP 550 Shinwod Dental Korea | Stereolithography, Layer height: 0.1 mm, Curing time: 3 sec | ODS C&B ODS CO Korea |

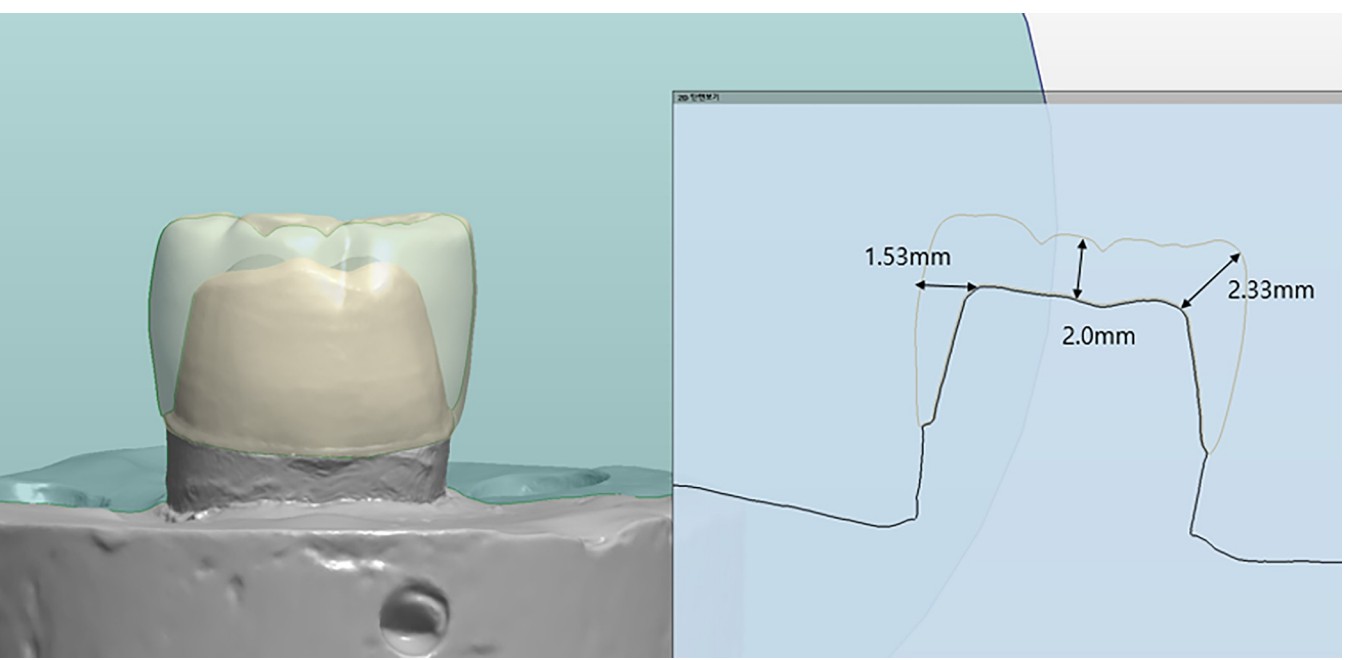

**Fig 1. Dimensions of the crown sample.**

dwell time of 30 s and a transition time of 10 s via a thermocycling machine (Thermal Cyclic Tester RB 508, R&B Inc., Korea) to simulate one month of the oral environment [21,22]. The sample size of 25 interim crowns for each group was determined via a sensitivity power analysis with 80% power, a 5% significance level [14] and an effect size of 0.4 [23] usingvia a software program (G*Power, v3.1.9.2; Heinrich-Heine-Universität Düsseldorf).

## Fracture strength and fracture mode

FS was measured after the thermocycling process. The test was conducted with a universal testing machine (Instron 3366; Instron Corporation). The samples cemented on each individual die using temporary cement (Kerr Dental, Brea, CA, USA) were placed on a holding jig, and a vertical load pressure was applied on the center of the samples with a 10 kN load cell at a crosshead speed of 1.0 mm/min using a 9.5 mm diameter steel ball until the samples fractured. The FS values were recorded in Newtons (N).

## Shore D hardness

The Shore D hardness was measured in both the before-thermocycling and after-thermocycling subgroups. Five measurements were performed at 25°C for each sample according to ISO 868 by placing the sample under the indenter of a Shore durometer (HPSD; Schmidt), and the mean value was recorded.

## Surface roughness

Surface roughness was measured in both the before-thermocycling and the after-thermocycling subgroups. The surfaces of two samples from each group were analyzed at 9 locations per sample via a 3D optical surface roughness analyzer with a vertical resolution of 0.05 nm and a root mean square (RMS) repeatability of 0.01 nm (Contour GT-X3 BASE; Bruker). The centerline

average roughness (Ra) and ten-point median height (Rz) were calculated. The objective magnification was 50×, and the zoom was 2×. The size of the field of view was 0.09 × 0.066 mm$^2$.

### Scanning electron microscopy (SEM)

To assess the surface topography, one sample per group was observed with field emission SEM (FE-SEM) (JEOL-7800F; JEOL, Ltd.) at an acceleration voltage of 2 kV and magnifications of x100 and x5000. The sample from each material group was left to dry at room temperature for 24 hours and then sputter-coated with gold and palladium for 180 s before FE–SEM examination.

### Statistical analysis

All the statistical analyses were performed using the SPSS 20 Statistics package (IBM SPSS; IBM Corp.) and reviewed by an independent statistician. Descriptive analysis was performed, and normality tests were conducted using Shapiro–Wilk test. To compare fracture strength after thermocycling and surface roughness before and after thermocycling, an independent t test was performed. Analysis of variance (ANOVA) was conducted to check for significant differences among the test groups, and the Bonferroni post hoc correction was used for multiple comparisons between individual groups. For Shore D hardness, the Wilcoxon rank sum test was performed due to the small sample size (N = 5) per group. A significance level of 0.05 was set for all statistical analyses.

## Results

### Fracture strength and fracture modes

The FS was the highest in the PMMA group (2787.93 N), followed by the bisphenol (2165.47 N) and PLA (2088.78 N) groups, but there was no significant difference between the bisphenol and PLA groups (Table 2).

The PMMA group predominantly exhibited vertical fractures directly downstream of the steel ball, but the fragments were retained without dislodgement. In the bisphenol group, midline fracture with a loss of more than half of the crown sample was mainly observed. The PLA group was characterized as close to tearing the material or deformation of the die instead of crown fracture (Fig 2).

### Shore D hardness

The highest Shore D hardness (HDS) before and after thermocycling was observed in the bisphenol group, followed by the PMMA and PLA groups (Table 3). While the Shore D hardness of each material in the PMMA and bisphenol groups did not significantly change after thermocycling ($p>0.05$), the PLA group showed a significant increase ($p<0.05$). The Shore D

**Table 2. Comparison of fracture strength.**

| Material | Fracture strength (N) |
|---|---|
| PLA | 2088.78±167.98[a] |
| PMMA | 2787.93±292.74[b] |
| Bisphenol | 2165.47±550.15[a] |

means±SDs; ANOVA (analysis of variance); the different letters indicate significant differences between groups according to the Bonferroni post hoc correction.

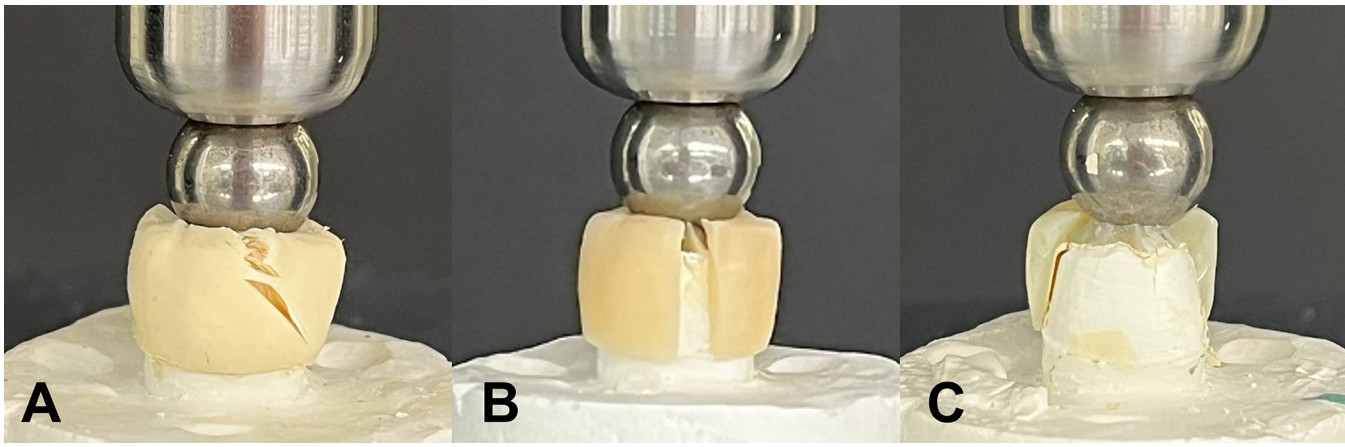

**Fig 2. Representative fracture pattern of each sample.** A. PLA, B. PMMA, and C. bisphenol groups.

hardness value decreased in the PMMA and bisphenol groups, resulting in insignificant differences between PMMA and bisphenol after the thermocycling procedure (Table 3).

## Surface roughness

In terms of Ra, which is a measure of the average roughness, the PMMA and PLA groups had similar Ra values, whereas the bisphenol group had the lowest roughness (Table 4, Fig 3). The 10-point median Rz increased in the order of the bisphenol, PMMA, and PLA groups. Within each material, the PMMA group showed no change in the surface roughness after thermocycling. For the PLA group, a significant increase in roughness was observed in Ra, but no change was observed in Rz. Notably, in the bisphenol group, significant increases in both Ra and Rz were observed after thermocycling, indicating that it was the roughest material among the three materials ($p<0.0001$).

## SEM observations

Representative FE-SEM images revealed that the PLA samples were generally smooth, but uniform irregularities in the form of layers due to filament stacking were observed (Fig 4A and 4A'). In the PMMA group, a uniform pattern due to the cutting process was observed, but the irregularities were less pronounced than those in the PLA group (Fig 4B and 4B'). The bisphenol samples were produced by photopolymerization, and no roughness in the pattern was observed (Fig 4C and 4C'). However, the bisphenol group showed a significant increase in roughness after thermocycling, which was confirmed by SEM images at 5000x (Fig 5C and 5C').

**Table 3. Comparison of Shore D hardness (HSD) values before and after thermocycling.**

| Material | Before (HSD) | After (HSD) | P* |
|---|---|---|---|
| PLA | 73.00(1.80)[a] | 77.40(0.80)[a] | 0.0472 |
| PMMA | 87.60(0.60)[b] | 85.80(6.20)[ab] | 0.0450 |
| Bisphenol | 90.60(0.40)[c] | 88.60(2.60)[b] | 0.2390 |

median (IQR); Kruskal–Wallis test; The different letters indicate significant differences between groups according to the Bonferroni post hoc analysis. *Wilcoxon rank sum test.

**Table 4. Comparison of surface roughness before and after thermocycling.**

| Material | Ra (μm) | | | Rz (μm) | | |
|---|---|---|---|---|---|---|
| | Before | After | *P** | Before | After | *P** |
| PLA | 0.44±0.07[a] | 0.54±0.14[a] | 0.0155 | 8.19±1.61[a] | 8.70±1.23[a] | 0.2940 |
| PMMA | 0.43±0.13[a] | 0.38±0.23[a] | 0.3556 | 5.97±1.34[b] | 5.43±2.01[b] | 0.3474 |
| Bisphenol | 0.28±0.08[b] | 1.02±0.41[b] | < .0001 | 3.89±0.83[c] | 11.01±3.43[c] | < .0001 |

Means±SDs; ANOVA (analysis of variance); the different letters indicate significant differences between groups according to the Bonferroni post hoc correction.
*independent t test.

## Discussion

This study aimed to assess the mechanical properties of a PLA interim single FDP after thermocycling to simulate intraoral conditions. The initial hypothesis that the mechanical properties of PMMA, bisphenol, and PLA would not significantly differ after thermocycling was partially confirmed. The method of thermocycling to simulate the one month usage of interim crowns based on ISO 7405 refers to previous studies, in which the number of cycles was varied

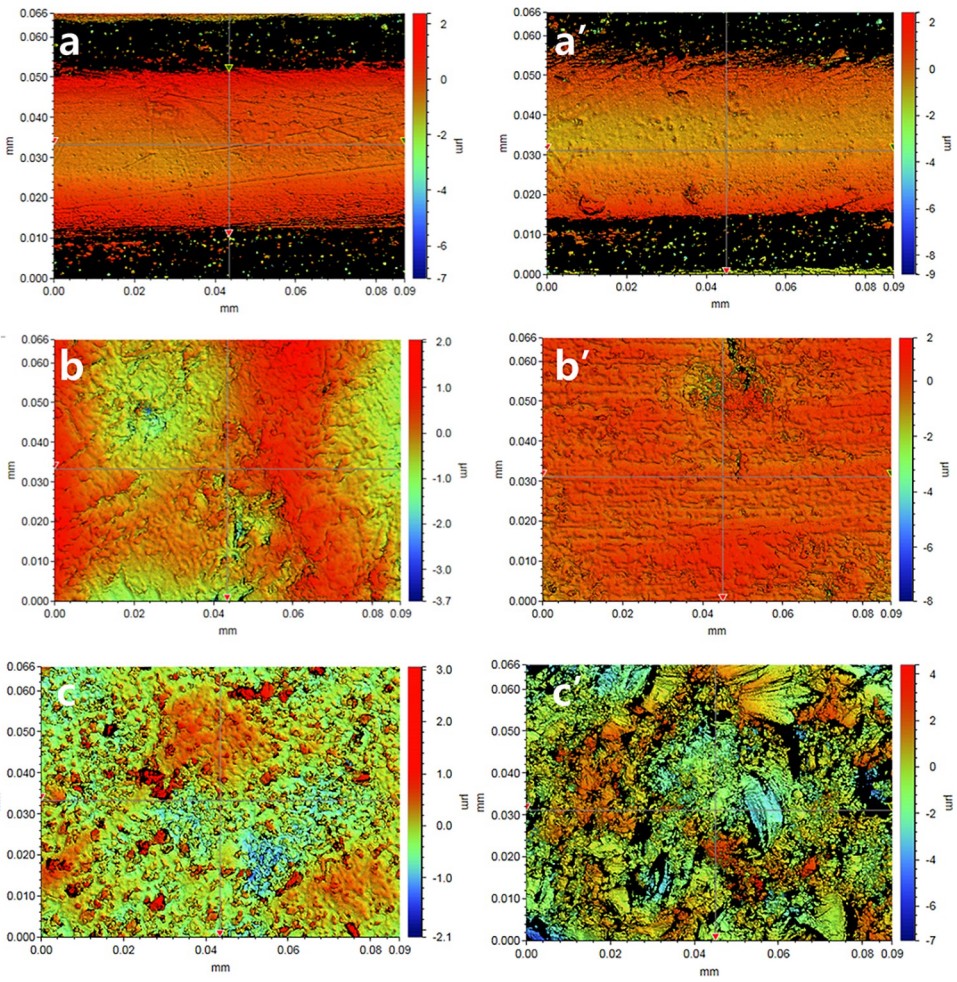

**Fig 3. Surface roughness (Ra) images of test groups.** (a) PLA; (a') PLA after thermocycling; (b) PMMA; (b') PMMA after thermocycling; (c) bisphenol; (c') bisphenol after thermocycling.

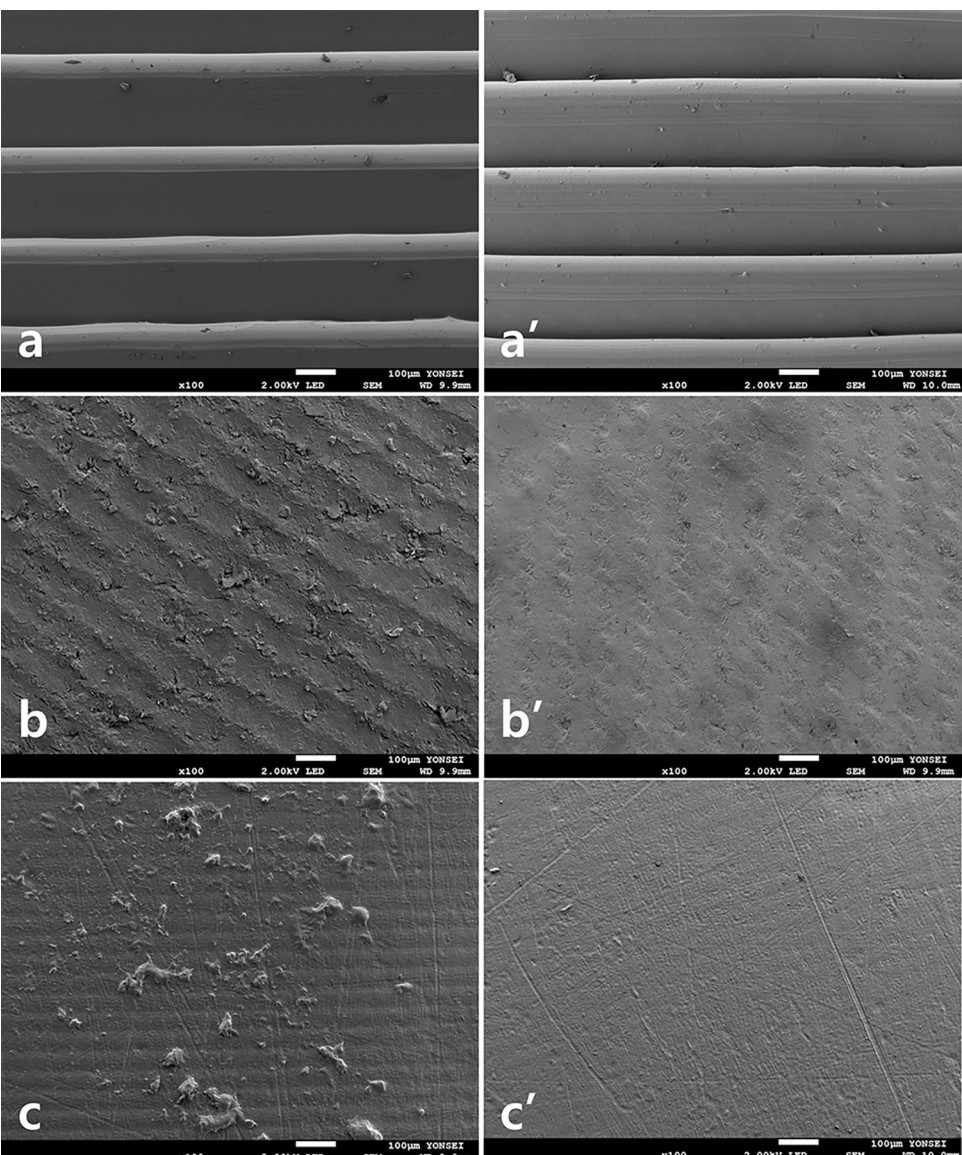

**Fig 4. Scanning electron microscopy (SEM) images of test groups at 100× magnification.** (a) PLA; (a') PLA after thermocycling; (b) PMMA; (b') PMMA after thermocycling; and (c) bisphenol; (c') bisphenol after thermocycling.

from 5000–60000 to simulate 6 months of use in the oral environment [14,22,24]. A total of 10000 thermocycles were performed to simulate 1-month usage in this study.

This study aimed to compare the mechanical properties of PLA with those of conventionally used CAD/CAM interim dental restorations. The parameters in this investigation were based on recommendations from previous studies and manufacturers [17–19]. For PLA, postprocessing annealing is the main process for improving mechanical properties; hence, the bed temperature is usually kept above the glass transition temperature (Tg, approximately 60°C) to maximize bonding between the deposited layers [25]. In this study, the nozzle temperature and bed temperature were maintained at approximately 200°C and 65°C, respectively, to maximize the balance between the degree of crystallinity and postprocessing annealing. Previous studies have explored the effects of the printing angle on the mechanical properties of 3D-

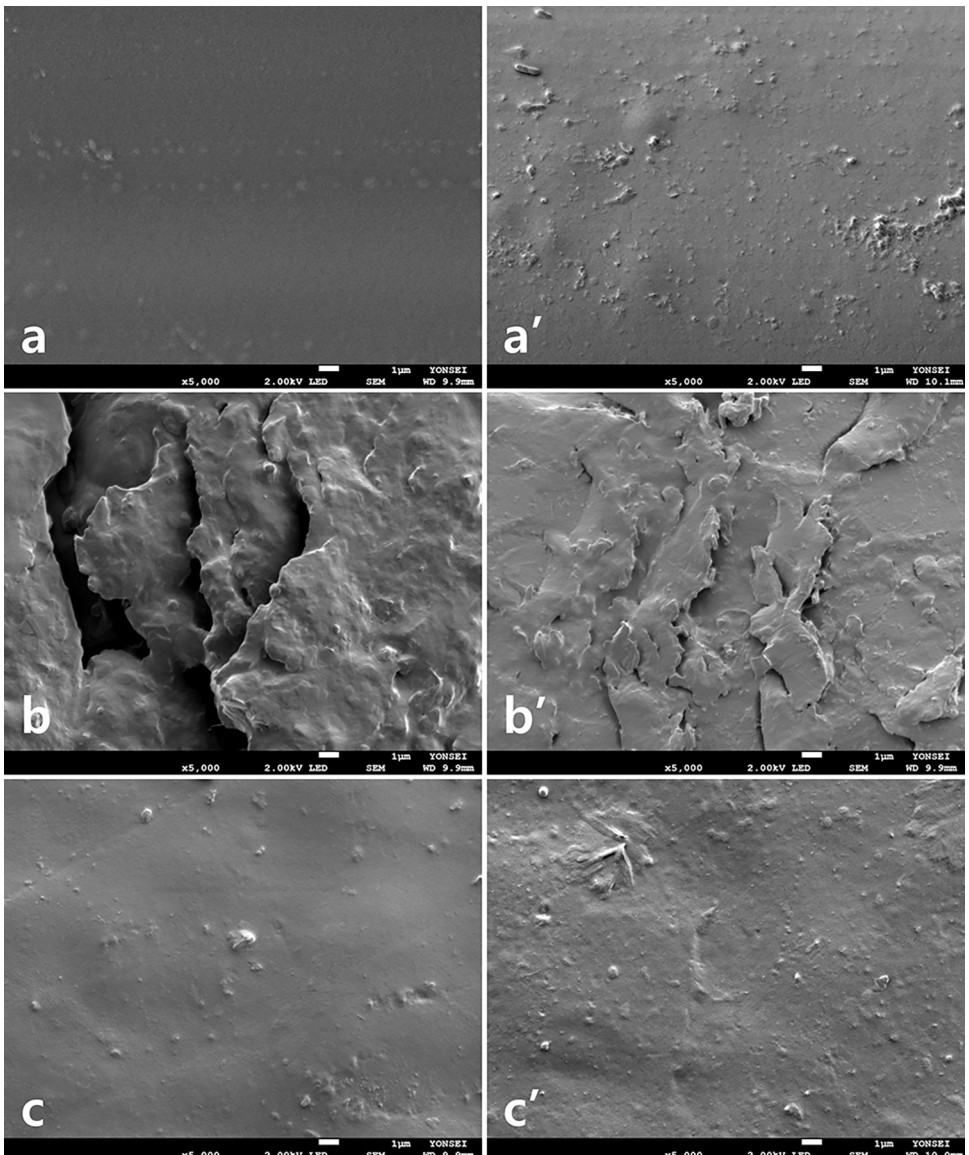

**Fig 5. Scanning electron microscopy (SEM) images of test groups at 5000× magnification.** (a) PLA; (a') PLA after thermocycling; (b) PMMA; (b') PMMA after thermocycling; and (c) bisphenol; (c') bisphenol after thermocycling.

printed interim dental restorations. Alharbi et al. [18] reported that a sample printed perpendicular to the load direction exhibited higher compressive strength, whereas Osman et al. [19] recommended 135° for DLP. Additionally, one study reported that printing at 30° resulted in the highest FS [17]. In this study, a 135-degree printing orientation for both PLA FDM and SLA was consistently used, based on prior research indicating that this angle was the ideal orientation for producing optimal mechanical properties.

In this study, the PMMA group presented the highest fracture resistance (2787.78 N) among the three groups, whereas the PLA (2088.78 N) and bisphenol (2165.47 N) groups showed similar strengths without significant differences. These values recorded in units of N were much greater than those reported in previous studies, in which the strengths of single- or 3-unit FDPs ranged from 540 N to 1350 N, depending on the materials used [14,16,21]. The

large difference in the absolute values of the data is thought to result from the sample design, test method, and CAD/CAM manufacturing parameters. Generally, the molecular alignment, weight, crystallinity, and postprocessing annealing of a printed polymer are affected by printing parameters, including the temperature, output position, build angle, number of layers, and configuration of the support structure [25,26]. The average maximum bite force is reported to vary widely between 286 and 727 N, ranging from 250–286 N for the anterior teeth and 580–727 N for the posterior teeth [27–30]. However, compared with that of the PMMA group produced via subtractive manufacturing and bisphenol group produced via SLA additive manufacturing, the FS of the PLA group produced via FDM additive manufacturing after thermocycling could be acceptable for clinical use.

The fracture pattern in the PLA group differed from that in the other groups, showing a torn pattern instead of the fractures or cracks observed in the PMMA group or bisphenol group, which was similar to the results of a previous study reporting that conventional PMMA, DLP-PMMA, and SLA-PMMA exhibited crack or fracture patterns; however, the flexural strength of the PLA group was difficult to measure because the samples deformed without breaking [17]. Our previous study reported that the elastic modulus of the PLA group was greater than those of the PMMA and bisphenol groups. This might be related to the fracture pattern of the PLA group [15]. During practical chewing, the ability of the PLA product, as an interim FDP, to deform rather than fracture and fail under functional loads may be advantageous, as long as the functional load is within the sustainable range of the PLA FDP.

The fracture pattern in each group can also be explained by the Shore D hardness results. In this study, the bisphenol group presented the highest value, followed by the PMMA and PLA groups, regardless of the thermocycling procedure, which was the same as the results of our previous study [15]. It could be inferred that the high Shore D hardness of the bisphenol group causes destructive fracture patterns and dislodgement when forces are concentrated at the occlusal contact. In contrast, the PLA group with a lower Shore D hardness was more likely to deform than to fail or fracture.

One of the main issues is whether the mechanical properties of an interim crown made of PLA are maintained over a provisional period in the humid environment of the oral cavity. How do temperature changes in the oral cavity with humidity affect the mechanical properties of PLA materials?

PLA degrades through hydrolysis of the backbone ester groups, and the rate of degradation depends on the crystallinity, molecular weight and distribution, morphology, water diffusion rate, and stereoisomer content of PLA. Because PLA is a hydrophobic and aliphatic polyester, the initial hydrolysis rate at the end of the polymer chain is very slow. More than 90% of the material has been reported to remain after 133 days at 37°C and after 28 days at 60°C [11]. Furthermore, hydrolysis is rapidly accelerated when carboxyl groups are formed at the end of the chain, forming water-soluble oligomers [11]. In the present study, the Shore D hardness in the PLA group increased after thermocycling, although the difference was not statistically significant. These findings suggest that PLA can maintain its molecular structure without undergoing hydrolysis even in a humid intraoral environment during the provisional function period. In addition, the Shore D hardness significantly differed between each group before thermocycling, but after thermocycling, a significant difference was observed only between the PLA group and the bisphenol group due to the increased value of the PLA group and the decreased values of the PMMA and bisphenol groups, which could further support the clinical potential of the PLA interim crown (Table 3).

Surface roughness is one of the important factors to consider for provisional restorative materials, as excessive increases in surface roughness in the intraoral environment could lead to concerns such as plaque accumulation, particularly for materials such as PLA with

hydrolytic properties. Notably, the bisphenol group presented the lowest surface roughness values in terms of Ra and Rz before thermocycling, but after thermocycling, it presented the highest values, with statistically significant differences among the experimental groups. The Ra values of the PLA group before and after thermocycling significantly differed but were not significantly different from those of PMMA. The Rz values before and after thermocycling were not significantly different.

Regarding bacterial adhesion, some *in vivo* studies have indicated a threshold surface roughness for bacterial retention (Ra = 0.2 μm), above which plaque accumulation significantly increases, heightening the risk of caries and periodontal inflammation. Based on this, the bisphenol group may not be ideal for long-term provisional FDP, while PLA could serve as a suitable alternative material for an interim prosthesis [31]. Therefore, bisphenol may not be ideal for a long-term provisional FDP, but PLA could serve as a suitable alternative material for an interim prosthesis.

The FE-SEM analysis and surface roughness results were not mutually supportive, even though the FE-SEM images with the mean surface roughness values among the test groups were selected (Table 4, Figs 4 and 5). This discrepancy is due to different sites being analyzed in the FE-SEM and surface roughness tests.

A few limitations of the present investigation should be addressed. This in vitro study could not reflect the complicated and diverse conditions of the oral cavity. In addition, even when an identical 3D printer or milling device is used to manufacture a provisional prosthesis, the mechanical properties of an interim prosthesis vary depending on the related parameters or conditions. Along with the technical improvement of FDM to increase the accuracy of PLA products and further research on PLA materials, such as their flexibility, conducting clinical trials is recommended to expand the analysis of the mechanical properties of PLA interim FDPs produced by additive manufacturing, with a focus on aspects such as biocompatibility, color stability, and reparability.

## Conclusion

Within the limitations of this in vitro study, the following conclusions were drawn:

1. After the thermocycling process, the PLA group produced via FDM additive manufacturing showed fracture strength, Shore D hardness, and surface roughness similar to those of the PMMA group produced via subtractive manufacturing and the bisphenol group produced via SLA additive manufacturing.

2. The PLA single interim FDP printed via FDM manufacturing maintained the appropriate mechanical properties after thermocycling, simulating a one-month provisional period; thus, PLA could be used as an alternative to conventional interim restoration materials.

## Supporting information

**S1 File.**
(XLSX)

**S2 File.**
(XLSX)

**S3 File.**
(XLSX)

## Author Contributions

**Conceptualization:** Bock-Young Jung.

**Data curation:** Seo Young Kim.

**Funding acquisition:** Bock-Young Jung.

**Investigation:** Re-Mee Doh, Won-Il Choi, Seo Young Kim.

**Methodology:** Won-Il Choi, Seo Young Kim.

**Resources:** Seo Young Kim.

**Validation:** Re-Mee Doh, Bock-Young Jung.

**Writing – original draft:** Re-Mee Doh.

**Writing – review & editing:** Bock-Young Jung.

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
