## [Decision Letter · Decision Letter 0]

5 Nov 2024

PONE-D-24-44087Mechanical properties of a polylactic 3D-printed interim crown after thermocyclingPLOS ONE

Dear Dr. Jung,

Thank you for submitting your manuscript to PLOS ONE. After careful consideration, we feel that it has merit but does not fully meet PLOS ONE’s publication criteria as it currently stands. Therefore, we invite you to submit a revised version of the manuscript that addresses the points raised during the review process.  Please submit your revised manuscript by Dec 20 2024 11:59PM. If you will need more time than this to complete your revisions, please reply to this message or contact the journal office at plosone@plos.org. Please include the following items when submitting your revised manuscript:A rebuttal letter that responds to each point raised by the academic editor and reviewer(s). You should upload this letter as a separate file labeled 'Response to Reviewers'.A marked-up copy of your manuscript that highlights changes made to the original version. You should upload this as a separate file labeled 'Revised Manuscript with Track Changes'.An unmarked version of your revised paper without tracked changes. You should upload this as a separate file labeled 'Manuscript'.

We look forward to receiving your revised manuscript.

Kind regards,

Esra Cengiz Yanardag

Academic Editor

PLOS ONE

3. Thank you for stating the following financial disclosure: “This work was supported by a faculty research grant from the Yonsei University College of Dentistry (No. 6-2023-0015).”

4. In the online submission form, you indicated that “The datasets used and/or analyzed during the current study are available from the corresponding author upon reasonable request.”

All PLOS journals now require all data underlying the findings described in their manuscript to be freely available to other researchers, either 1. In a public repository, 2. Within the manuscript itself, or 3. Uploaded as supplementary information. This policy applies to all data except where public deposition would breach compliance with the protocol approved by your research ethics board. If your data cannot be made publicly available for ethical or legal reasons (e.g., public availability would compromise patient privacy), please explain your reasons on resubmission and your exemption request will be escalated for approval.

Reviewers' comments:

**Comments to the Author**

1. Is the manuscript technically sound, and do the data support the conclusions?

Reviewer #1: Partly

Reviewer #2: Yes

2. Has the statistical analysis been performed appropriately and rigorously? 

Reviewer #1: Yes

Reviewer #2: I Don't Know

3. Have the authors made all data underlying the findings in their manuscript fully available?

Reviewer #1: Yes

Reviewer #2: Yes

4. Is the manuscript presented in an intelligible fashion and written in standard English?

Reviewer #1: Yes

Reviewer #2: Yes

5. Review Comments to the Author

Reviewer #1: The manuscript is interesting. It offers information on current material, obtained using increasingly used technology, and may contribute to the field.

However, some considerations need to be made:

-Introduction

You address, at some point, the importance of the parameters used in each method, but nothing is said about it in the Introduction or in Materials and Methods.

Materials and Methods

- In the fracture toughness test, isn't the loading speed at 1.00mm/min a little high? Why was this speed used?

- What parameters are used in each method of acquisition? Explain them all!

Results

The images in Figures 3, 4 and 5 are of low quality and details end up not being visible. Furthermore, they were little explored.

Discussion

When you quote the fracture toughness value for PMMA, you can also quote those found for other materials, making it easier to read.

You state that "Generally, the molecular alignment, weight, crystallinity, and postprocessing annealing of a printed polymer are affected by printing parameters, including the temperature, output position, build angle, number of layers, and configuration of the support structure [22,23]." What is the relationship between the parameters you used and the results found? This needs to be explored!

You claim that the flexural strength of PLA is difficult to measure because the material deforms without breaking. However, after a certain deformation the material becomes unusable, so couldn't it be established how much deformation would make it unusable?

And in what way can this deformation capacity really be an advantage?

IF PLA is said to have a high modulus of elasticity, wouldn't it be a rigid material, therefore with a brittle fracture characteristic?

You have researched the roughness of materials. What was the reason that led you to do this? Explore the importance of this test in clinical use!!!

Regarding the limitations pointed out for the study, do you really think that's all there is to it? If they were to do the study today, would nothing change?

Reviewer #2: The manuscript is interesting, and the methodology is well-conducted. I have some suggestions aimed at improving the manuscript.

Introduction

It would be interesting to add references to support the first paragraph.

The second paragraph consists of only one sentence, which is too long. Scientific writing often favors short sentences. Please restructure the second paragraph by using periods to break up the sentences.

The sentence about the aim of the study is too long. It makes the sentence quite unclear. Please write more objectively, with shorter sentences. I suggest removing 'to mimic the intraoral environment and evaluate the potential for clinical application”. In addition, thermocycling does not evaluate the potential for clinical application. It is merely an aging methodology.

Materials and Methods

In the scanning electron microscopy analysis, it would be relevant to inform the magnification used to observe the surface topography.

It is important to clarify that shore D hardness and surface roughness measurements were taken before and after thermocycling. Please add this information to the text.

In relation to the surface roughness methodology, I would like to know if the measurements taken on the sample, before and after thermocycling, were conducted in the same direction on the material’s surface.

Result

What were the p values for the Shapiro-Wilk test? Did the data show normal distribution?"

The data on fracture strength, Shore D hardness, and surface roughness are numerical outcomes. Therefore, a parametric test would be expected. However, non-parametric tests were applied. What happened with the data?

It reads “...Wilcoxon signed rank test was performed to compare before and after thermocycling within the groups”. This sentence is not clear. I assume that measurements before and after thermocycling were taken for the Shore D hardness and surface roughness, since fracture strength test is a destructive method. Please clarify the information.

It reads “Within each material, the PMMA group showed no change in the surface roughness after heat treatment”. Please change 'heat treatment' to 'thermocycling.

Discussion

In general, the discussion is well-written and justifies the results found in the study. However, regarding surface roughness, it would be helpful to add information about the threshold of 0.2 micrometers, as this value is related to bacterial adhesion on the material’s surface.

6. PLOS authors have the option to publish the peer review history of their article (what does this mean?). If published, this will include your full peer review and any attached files.

Reviewer #1: No

Reviewer #2: No

---

## [Author Response · Author response to Decision Letter 0]

9 Dec 2024

Reviewer #1: The manuscript is interesting. It offers information on current material, obtained using increasingly used technology, and may contribute to the field. However, some considerations need to be made:

Introduction

1. You address, at some point, the importance of the parameters used in each method, but nothing is said about it in the Introduction or in Materials and Methods.

Response: We appreciate this comment.

The aim of this study was to compare the mechanical properties of PLA with those of conventionally used CAD/CAM interim dental restorations. The parameters applied in this investigation were the values that have been recommended in previous studies or by the manufacturers. We provide the parameters used for each CAD/CAM method in Table 1. For PLA FDM, the parameters used are consistent with those in our previous study (Mechanical properties of CAD/CAM polylactic acid as a material for interim restoration, Heliyon 2023).

Text changed: See the section of ‘Material and methods”

The parameters applied to this investigation were the values that have been recommended in previous studies or by the manufacturers, including the layer height, nozzle size, ejection speed, and curing time (Table 1).

Materials and Methods

2. In the fracture toughness test, isn't the loading speed at 1.00mm/min a little high? Why was this speed used?

Response: Thank you for the comments.

Generally, the physical or mechanical properties of dental polymers are tested following ISO 10477 standards. In addition, a head speed of 1.00 mm/min was used in several previous studies on CAD/CAM dental polymers. Accordingly, our tests used parameters based on ISO 10477 (2020) to ensure the reliability of the data used for measuring polymer performance and to compare them with those used in previous studies. According to the literature, the head speed of the Instron machine for fracture strength testing varies slightly across studies.

For your reference, we have directly quoted the descriptions from the materials and methods sections of the studies we consulted:

• Reeponmaha et al., “All cemented provisional crowns were subjected to the universal testing machine (Lloyd LR10K, Ametek, FL, USA) under an axial load at a crosshead speed of 1 mm/min with 30 kN load cells, using a metal ball of 5 mm diameter at the central pit parallel to the long axis of the tooth until failure occurred.”

• Martín-Ortega et al., “A universal testing machine (BT1-FR2.5TS. D14; Zwick Roell) was used for the fracture resistance analysis. The load was applied at a speed of 1 mm/min until fracture occurred.”

References:

T. Reeponmaha et al, Comparison of fracture strength after thermos-mechanical aging between provisional crowns made with CAD/CAM and conventional method. J Adv Prosthodont 2020; 12:218-24.

N. Martín-Ortega et al. Fracture resistance of additive manufactured and milled implant-supported interim crowns. J Prosthet Dent 2022; 127:267-74.

Text changed: N‒S

3. What parameters are used in each method of acquisition? Explain them all!

Response: Thank you for the comments.

We have made every effort to detail the parameters used in this study through both the text and accompanying tables. We summarized the specific parameters utilized as follows:

FDM Method

• Printer: KPLA (3D KOREA CO., Korea)

• Manufacturing Parameters (preset recommendation):

o Filament diameter (Φ): 1.75 mm

o Layer height: 0.1 mm

o Nozzle size: 0.4 mm

o Printing speed: 60 m/s

o Nozzle temperature: 200 °C

o Bed temperature: 65 °C

• Posttreatment: None required

SLA Method

• Printer: ODS C&B (ODS CO., Korea)

• Manufacturing Parameters (recommended by the manufacturer):

o Layer height: 0.1 mm

o Curing time: 3 seconds

• Posttreatment:

1. The samples were immersed in 100% isopropyl alcohol (Medifive, Tornado, Korea) to remove the resin monomers.

2. Postpolymerization was conducted for 210 seconds with a UV light polymerization unit (LC-3D Print Box, NextDent, Netherlands).

Text changed: See Table 1, “Materials and methods”, and “Discussion”.

Results

4. The images in Figures 3, 4 and 5 are of low quality and details end up not being visible. Furthermore, they were little explored.

Response: We appreciate these comments.

The current images in Figures 3, 4 and 5 were set at 330 DPI (dots per inch), which follow the PLOS ONE guidelines that recommend a DPI range of 300-600. We have tried to improve the image quality by using 600 DPI.

The images were changed: See “Figures 3, 4, and 5.”

Discussion

5. When you quote the fracture toughness value for PMMA, you can also quote those found for other materials, making it easier to read.

Response: Thank you for your detailed comments.

Text changed: We added the fracture toughness values of the PLA and bisphenol groups to the “Discussion.”

6. You state that "Generally, the molecular alignment, weight, crystallinity, and postprocessing annealing of a printed polymer are affected by printing parameters, including the temperature, output position, build angle, number of layers, and configuration of the support structure [22,23]." What is the relationship between the parameters you used and the results found? This needs to be explored!

Response: We appreciate this comment.

This study aimed to compare the mechanical properties of PLA with those of conventionally used CAD/CAM interim dental restorations. The parameters used in this investigation were based on recommendations from previous studies and manufacturers. Future research could explore the relationships between these parameters and the mechanical properties of PLA. We have included details regarding the specified printing parameters in the text as follows.

Test changed: See “Discussion”.

This study aimed to compare the mechanical properties of PLA with those of conventionally used CAD/CAM interim dental restorations. The parameters in this investigation were based on recommendations from previous studies and manufacturers. For PLA, postprocessing annealing is the main process for improving mechanical properties; hence, the bed temperature is usually kept above the glass transition temperature (Tg, approximately 60°C) to maximize bonding between the deposited layers[23]. In this study, the nozzle temperature and bed temperature were maintained at approximately 200°C and 65°C, respectively, to maximize the balance between the degree of crystallinity and postprocessing annealing. Previous studies have explored the effects of the printing angle on the mechanical properties of 3D-printed interim dental restorations. Alharbi et al. reported that a sample printed perpendicular to the load direction exhibited higher compressive strength, whereas Osman et al. recommended 135° for DLP. Additionally, one study reported that printing at 30° resulted in the highest FS. In this study, a 135-degree printing orientation for both PLA FDM and SLA was consistently used, based on prior research indicating that this angle was the ideal orientation for producing optimal mechanical properties[17, 24, 25]. 

References:

Alharbi et al, Effects of build direction o the mechanical properties fo 3D printed complete coverage interim dental restorations, J. Prosthet. Dent 115(6) (2016) 760-767

Park et al, Flexural strength of 3D printing resin materials ofr provisional FDP, Materials13(18)(2020)

Osman et al, Build angle: does it ingluence the accuracy of 3D pinted dental restorations using DLP technology? Int. J. Prosthodont(IJP) 30 (2) (2017) 182-188.

7. You claim that the flexural strength of PLA is difficult to measure because the material deforms without breaking. However, after a certain deformation the material becomes unusable, so couldn't it be established how much deformation would make it unusable? And in what way can this deformation capacity really be an advantage? 

Response: We appreciate this keen opinion from the reviewer. We have discussed this issue with the experts of the Department of Dental Materials. From the dental prosthodontic point of view, marginal sealing achieved by cement should be maintained under any circumstances or functional loads to protect a target tooth while maintaining the structural integrity of an interim dental prosthesis. Therefore, testing to assess the flexibility of a material while maintaining a seal is recommended to determine how much deformation would make it unusable and how flexible the material is.

This issue is a limitation of this study and an interesting theme for future investigations.

Text changed: N‒S

8. IF PLA is said to have a high modulus of elasticity, wouldn't it be a rigid material, therefore with a brittle fracture characteristic?

Response: We appreciate this valuable comment.

In our previous study, although PLA is a nonbrittle and flexible material, the resulting value of EM was the highest because it is based on the fracture load, and the PLA material has flexibility.

The flexural strength and elastic modulus values of PLA were calculated by transforming the maximum fracture value (N) with the following formulas:

for FS, σf= (3 × P × L)/(2 × b × h2)

for EM, Ef=(L3 × m)/(4 × b × h3)

P: maximal load (N), L: support span (mm), b: width of the specimen at the failure site (mm), h: height of the specimen at the failure site, and m: gradient of the initial straight-line portion of the load‒deflection curve (N/mm).

The study data could explain or infer the subsequent results; therefore, we have changed the phrase in the text “Our previous study reported that the elastic modulus of the PLA specimens was greater than those of the PMMA and bisphenol specimens. This could explain the fracture pattern of PLA .” to “Our previous study reported that the elastic modulus of the PLA specimens was greater than those of the PMMA and bisphenol specimens. This might be related to the fracture pattern of the PLA[15].”

Text changed: see the “Discussion” section.

9. You have researched the roughness of materials. What was the reason that led you to do this? Explore the importance of this test in clinical use!!!

Response: We appreciate your recommendation.

Given that PLA has self-hydrolyzing properties, we conducted this study because excessive increases in surface roughness in the intraoral environment could lead to concerns such as plaque accumulation. Assessing these changes and comparing them to those observed in materials currently in use holds significant clinical relevance, as it allows us to evaluate the suitability of PLA for provisional restorations. Additionally, we have clarified the text in the manuscript to specify the subject more clearly regarding the need for surface roughness assessment and added a reference that showed a relationship between surface roughness and bacterial retention (Bollen et al. Dent Mater 1997).

Text Changed:

Surface roughness is one of the important factors to consider for provisional restorative materials and could be vulnerable to changes in temperature and humidity in the intraoral environment.

-> Surface roughness is one of the important factors to consider for provisional restorative materials, as excessive increases in surface roughness in the intraoral environment could lead to concerns such as plaque accumulation, particularly for materials such as PLA with hydrolytic properties.

Text added:

Regarding bacterial adhesion, some in vivo studies have indicated a threshold surface roughness for bacterial retention (Ra = 0.2 μm), above which plaque accumulation significantly increases, heightening the risk of caries and periodontal inflammation. Based on this, the bisphenol group may not be ideal for long-term provisional FDP, while PLA could serve as a suitable alternative material for an interim prosthesis[31]. 

Reference: Curd M. L. Bollen et al. Comparison of surface roughness of oral hard materials to the threshold surface roughness for bacterial plaque retention: A review of the literature. Dent Mater 13:258-269, 1997

10. Regarding the limitations pointed out for the study, do you really think that's all there is to it? If they were to do the study today, would nothing change?

Response: Thank you for considering the limitations of this study.

Several confounding factors could challenge the data reliability, such as time, environment, and technicians. Because methodological complications might arise unexpectedly, a pilot study could help researchers avoid these difficulties and improve the data reliability. Our previous study (15) served as a pilot study. Nevertheless, several issues remain to be addressed in future research:

1. Is PLA flexibility beneficial or not?

In what way can this deformation capacity be advantageous?

2. How can the EM of the PLA be determined?

3. Which parameters of FDM could be used for establishing the accuracy of PLA products?

Text Changed: see the “Discussion” section.

Along with the technical improvement of FDM to increase the accuracy of PLA products and further research on PLA materials, such as their flexibility, conducting clinical trials is recommended to expand the analysis of the mechanical properties of PLA interim FDPs produced by additive manufacturing, with a focus on aspects such as biocompatibility, color stability, and reparability. 

Reviewer #2: The manuscript is interesting, and the methodology is well-conducted. I have some suggestions aimed at improving the manuscript.

Introduction

1. It would be interesting to add references to support the first paragraph.

Response: Thank you for the comments.

We have added the following reference to support this paragraph:

[1] Frederick, D.R., 1975. The provisional fixed partial denture. J. Prosthet. Dent. 34

2. The second paragraph consists of only one sentence, which is too long. Scientific writing often favors short sentences. Please restructure the second paragraph by using periods to break up the sentences.

Response: Thank you for this comment.

Following your comment, we have separated the sentences to clarify their meaning.

Text Changed:

Dental polymers primarily used as interim materials are limited to biomaterials such as polymethylmethacrylate (PMMA), bisphenol resins, and polyacrylate-ethene (PAEK). Although each material has certain limitations, such as high shrinkage of PMMA and low flowability of bisphenol A-glycidyl methacrylate (Bis-GMA) and urethane dimethacrylate (UDMA), they have been optimized for use in dental applications, minimizing the impact of these weaknesses[2].

3. The sentence about the aim of the study is too long. It makes the sentence quite unclear. Please write more objectively, with shorter sentences. I suggest removing 'to mimic the intraoral environment and evaluate the potential for clinical application”. In addition, thermocycling does not evaluate the potential for clinical application. It is merely an aging methodology.

Response: We appreciate your recommendation.

We have rephrased the sentence concerning the aim of the study to be clear. Additionally, we have modified the description of thermocycling accordingly.

Text Changed:

Therefore, this study aimed to evaluate the potential clinical use of PLA as a material for interim crowns by comparing its mechanical properties, such as fracture strength (FS), Shore D hardness, and surface roughness, with those of conventionally used CAD/CAM dental polymers, specifically PMMA (via subtractive manufacturing) and bisphenol (via SLA). Thermocycling was performed to replicate aging in the intraoral environment. The hypothesis was that the mechanical properties of PMMA, bisphenol, and PLA would not significantly differ after thermocycling.

Materials and Methods

4. In the scanning electron microscopy analysis, it would be relevant to inform the magnification used to observe the surface topography.

Response: We appreciate this vital recommendation.

Following the reviewer’s suggestion, we have added the magnification used in the scanning electron microscopy analysis.

Text Changed:

To assess the surface topography, one sample per group was observed with field emission SEM (FE-SEM) (JEOL-7800F; JEOL, Ltd.) at an acceleration voltage of 2 kV and magnifications of x100 and x5000. 

5. It is important to clarify that shore D hardness and surface roughness measurements were taken before and after thermocycling. Please a

---

## [Decision Letter · Decision Letter 1]

5 Jan 2025

PONE-D-24-44087R1Mechanical properties of a polylactic 3D-printed interim crown after thermocycling

Please present reference(s) for every sentence in introduction section and also in discussion section.

Please add reference(s) for this sentence: The parameters applied to this investigation were the values that have been recommended in previous studies or by the manufacturers, including the layer height, nozzle size, ejection speed, and curing time (Table 1)

The paragraph in line 249-261 needs references. You mentioned ‘’previous studies’’ however any studies were cited. Please include citations for Alharbi, Osman, and the study recommending 30° printing orientation.

We look forward to receiving your revised manuscript.

Kind regards,

Esra Cengiz-Yanardag

Academic Editor

PLOS ONE

Journal Requirements:

Additional Editor Comments:

Please present reference(s) for every sentence in introduction section and also in discussion section.

Please add reference(s) for this sentence: The parameters applied to this investigation were the values that have been recommended in previous studies or by the manufacturers, including the layer height, nozzle size, ejection speed, and curing time (Table 1)

The paragraph in line 249-261 needs references. You mentioned ‘’previous studies’’ however any studies were cited. Please include citations for Alharbi, Osman, and the study recommending 30° printing orientation.

Reviewers' comments:

Reviewer's Responses to Questions

**Comments to the Author**

1. If the authors have adequately addressed your comments raised in a previous round of review and you feel that this manuscript is now acceptable for publication, you may indicate that here to bypass the “Comments to the Author” section, enter your conflict of interest statement in the “Confidential to Editor” section, and submit your "Accept" recommendation.

Reviewer #1: All comments have been addressed

Reviewer #2: All comments have been addressed

2. Is the manuscript technically sound, and do the data support the conclusions?

Reviewer #1: Yes

Reviewer #2: Yes

3. Has the statistical analysis been performed appropriately and rigorously? 

Reviewer #1: Yes

Reviewer #2: Yes

4. Have the authors made all data underlying the findings in their manuscript fully available?

Reviewer #1: Yes

Reviewer #2: Yes

5. Is the manuscript presented in an intelligible fashion and written in standard English?

Reviewer #1: Yes

Reviewer #2: Yes

6. Review Comments to the Author

Reviewer #1: I appreciate the answers to my questions. The effort was worth it and the manuscript is much better and more complete.

Reviewer #2: The authors have adequately addressed my comments and suggestions. The manuscript has improved. In my opinion, it is suitable for publication.

7. PLOS authors have the option to publish the peer review history of their article (what does this mean?). If published, this will include your full peer review and any attached files.

Reviewer #1: No

Reviewer #2: No

---

## [Author Response · Author response to Decision Letter 1]

7 Jan 2025

Comments to the Author

Reviewer #1: I appreciate the answers to my questions. The effort was worth it and the manuscript is much better and more complete.

Reviewer #2: The authors have adequately addressed my comments and suggestions. The manuscript has improved. In my opinion, it is suitable for publication.

Additional Editor Comments:

1. Please present reference(s) for every sentence in introduction section and also in discussion section.

Response: Thank you for the comments.

We have carefully reviewed the Introduction and Discussion sections to ensure that all external research findings and information, apart from the results of this study, have been appropriately cited with references.

2. Please add reference(s) for this sentence: The parameters applied to this investigation were the values that have been recommended in previous studies or by the manufacturers, including the layer height, nozzle size, ejection speed, and curing time (Table 1)

Response: Thank you for this comments.

We have added the references at their appropriate positions in the Introduction and Discussion sections as requested. No additional references were added. The reference numbers have been adjusted according to the citation order in the main text.

Text changed:

The parameters applied to this investigation were the values that have been recommended in previous studies [17-19] or by the manufacturers, including the layer height, nozzle size, ejection speed, and curing time (Table 1).

3. The paragraph in line 249-261 needs references. You mentioned ‘’previous studies’’ however any studies were cited. Please include citations for Alharbi, Osman, and the study recommending 30° printing orientation.

Response: Thank you for the comments.

To enhance clarity, we have now cited each reference individually at the specific points where they are mentioned. Consequently, the reference numbers have been adjusted to reflect the citation order in the main text.

Text changed:

This study aimed to compare the mechanical properties of PLA with those of conventionally used CAD/CAM interim dental restorations. The parameters in this investigation were based on recommendations from previous studies and manufacturers[17-19]. For PLA, postprocessing annealing is the main process for improving mechanical properties; hence, the bed temperature is usually kept above the glass transition temperature (Tg, approximately 60°C) to maximize bonding between the deposited layers[25]. In this study, the nozzle temperature and bed temperature were maintained at approximately 200°C and 65°C, respectively, to maximize the balance between the degree of crystallinity and postprocessing annealing. Previous studies have explored the effects of the printing angle on the mechanical properties of 3D-printed interim dental restorations. Alharbi et al.[18] reported that a sample printed perpendicular to the load direction exhibited higher compressive strength, whereas Osman et al.[19] recommended 135° for DLP. Additionally, one study reported that printing at 30° resulted in the highest FS [17]. In this study, a 135-degree printing orientation for both PLA FDM and SLA was consistently used, based on prior research indicating that this angle was the ideal orientation for producing optimal mechanical properties.

Thank you very much again.

We appreciate the reviewers’ comments, which have helped us to improve our manuscript.

Sincerely,

Bock Young Jung

---

## [Editor Report · Decision Letter 2]

12 Jan 2025

Mechanical properties of a polylactic 3D-printed interim crown after thermocycling

PONE-D-24-44087R2

Dear Dr. Jung

We’re pleased to inform you that your manuscript has been judged scientifically suitable for publication and will be formally accepted for publication once it meets all outstanding technical requirements.

Kind regards,

Esra Cengiz Yanardag

Academic Editor

PLOS ONE
---

## [Editor Report · Acceptance letter]

16 Jan 2025

PONE-D-24-44087R2 

PLOS ONE

Dear Dr. Jung, 

I'm pleased to inform you that your manuscript has been deemed suitable for publication in PLOS ONE. Congratulations! Your manuscript is now being handed over to our production team.

Kind regards, 

on behalf of

Dr. Esra Cengiz Yanardag 

Academic Editor

PLOS ONE